# Identification of active sites on supported metal catalysts with carbon nanotube hydrogen highways

Nicholas M. Briggs[1], Lawrence Barrett[1], Evan C. Wegener[2], Leidy V. Herrera[1], Laura A. Gomez[1], Jeffrey T. Miller[2] & Steven P. Crossley [1]

Catalysts consisting of metal particles supported on reducible oxides exhibit promising activity and selectivity for a variety of current and emerging industrial processes. Enhanced catalytic activity can arise from direct contact between the support and the metal or from metal-induced promoter effects on the oxide. Discovering the source of enhanced catalytic activity and selectivity is challenging, with conflicting arguments often presented based on indirect evidence. Here, we separate the metal from the support by a controlled distance while maintaining the ability to promote defects via the use of carbon nanotube hydrogen highways. As illustrative cases, we use this approach to show that the selective transformation of furfural to methylfuran over $Pd/TiO_2$ occurs at the $Pd-TiO_2$ interface while anisole conversion to phenol and cresol over $Cu/TiO_2$ is facilitated by exposed $Ti^{3+}$ cations on the support. This approach can be used to clarify many conflicting arguments in the literature.

[1] School of Chemical, Biological and Materials Engineering, University of Oklahoma, Norman, OK 73019, USA. [2] Davidson School of Chemical Engineering, Purdue University, 480 Stadium Mall Drive, West Lafayette, IN 47907, USA. Correspondence and requests for materials should be addressed to S.P.C. (email: stevencrossley@ou.edu)

Metals deposited on oxide supports have demonstrated promising activity for numerous reactions including C–C cleavage, C–O cleavage, and C–O oxidation reactions. While examples of superior catalytic activity and selectivity when combining two different catalytically active substances are abundant, the fundamental reason for this enhanced activity is often unknown or contradictory literature exists. Many of these active oxide supports are also reducible oxides, with OH groups and oxygen vacancies present on their surface under reaction conditions. The interaction of a reducible metal oxide with a metal capable of dissociating hydrogen can be complex due to the creation of several potential catalytically active sites under reduction conditions. These active sites can be separated into two general categories, short-range interactions and long-range interactions. Short-range interactions occur in close proximity to the metal particle and include the reduction of the interface along the perimeter of the metal, electronic perturbations to the exposed metal surface due to interaction with the support, as well as the formation of thin oxide films over the metal surface[1–4]. Surface defects, subsurface defects, and surface OH groups that are created by hydrogen spillover from the metal nanoparticle to the support can be considered long-range interactions, or promoter effects, as the primary active sites are found on the support itself[5].

Understanding the location and nature of the catalytic active site is critical for controlling a catalyst's activity and selectivity. A variety of methods have been applied to determine the active sites responsible for the activity observed over these catalysts. One approach is to change the particle size of the metal supported on the oxide, which changes the perimeter of the metal oxide interface[1–4]. By systematically modifying the perimeter and particle size, correlations with the observed rate can be developed between perimeter sites and sites on the metal with varying coordination. Lack of direct correlation of reaction rate with perimeter or specific metal surface sites are often associated with promoter effects. However, the metal oxide interface can change due to strong metal support interaction involving decoration or encapsulation of the metal particle by the reducible oxide support[1,6–8]. Spectroscopic techniques are also used to elucidate the nature of active sites. However, observable vibrations from spectator species that are often difficult to distinguish from catalytically relevant intermediates complicate this approach. These complications have led to a great deal of speculation pertaining to the location of the active site, often without any direct proof.

In this work, vertically grown carbon nanotubes are used to segregate direct catalytic interactions from promoter effects. This is accomplished by selective deposition of the metal and oxide catalysts by a precise distance through a conductive bridge of carbon nanotubes serving as hydrogen highways. A combination of probe reactions and characterization are used to validate this approach for discerning promoter effects from activity resulting from direct contact between the metal and the support. We use this approach to discern the kinetic relevance of these two families of potential active sites for the selective transformation of furfuraldehyde to methylfuran over a Pd/TiO$_2$ catalyst, as well as for the conversion of anisole to phenol and cresol over a Cu/TiO$_2$ catalyst. We confirm that the nanotubes are capable of serving as a shuttle to facilitate the generation of defects on the TiO$_2$ surface through a combination of temperature programmed reduction (TPR) techniques and X-ray absorption spectroscopy. We further use probe reactions that are sensitive to the number of exposed Ti$^{3+}$ cations to confirm that said defects on the TiO$_2$ surface are present under reaction conditions required for the selective C–O cleavage reactions in question.

## Results

**Synthesis and characterization.** Selective deposition of catalytic sites at a specific distance along the length of a nanotube was accomplished through use of vertically grown carbon nanotube forests. A metal evaporator was used to deposit metals such as Pd, Cu or Ti on the ends of the forest. A schematic illustrating the desired material obtained with this approach is shown in Fig. 1a. Subsequent calcination results in the formation of their respective oxides (e.g., TiO$_2$). This approach eliminates direct contact of the metal nanoparticle with the oxide support while maintaining the formation and regeneration of prospective active sites on the oxide. Promoter effects of the metal on the active support (e.g., TiO$_2$) are still possible in this scenario due to the fact that carbon nanotubes are known to facilitate spillover of dissociated hydrogen from the metal[9–13].

Loadings are determined through a quartz crystal microbalance. The amounts of catalyst deposited can be found in the Supplementary Table 1. Figure 1 shows energy-dispersive X-ray spectroscopy (EDS) spectra taken at various positions along the length of the nanotube forest where Pd metal was deposited on one end of the forest and TiO$_2$ on the other. Catalyst particles are known to sinter and migrate at various rates depending on the catalyst support[14,15]. To ensure that migration is not sufficient to induce significant physical contact, EDS spectra taken after 1 h of treatment at 400 °C in hydrogen confirm that the Pd and TiO$_2$ are not migrating across the length of the nanotube and the two catalysts are not coming into contact. No significant penetration of Pd, Cu, or TiO$_2$ is observed beyond five microns depth into the forest in any case. Target metal signals are within the Bremsstrahlung background levels in the center of the forest. In the case of separated metal and oxide, the forest was manipulated by removing from the silicon wafer via aluminum tape containing a carbon adhesive to position the forest for metal deposition on each side. Trace amounts of residue from the tape are removed via a mild thermal treatment in an oxidizing atmosphere.

**Evidence for hydrogen spillover.** TPR experiments are used to confirm the ability of nanotubes to serve as hydrogen highways connecting metals with oxides that are not in physical contact. It is well known that Pd can reduce and dissociate hydrogen at temperatures well below those required for dissociation on Cu. Pd catalysts can be reduced at 100 °C or even room temperature while CuO reduces just above 200 °C in atmospheric pressure hydrogen[16]. The TPR profile for Pd and CuO spatially separated on carbon nanotubes (Pd/CNT/CuO) is compared with identically deposited CuO on carbon nanotubes alone (CuO/CNT) in Fig. 2a. EDS spectra of this catalyst after high temperature reduction confirming lack of direct Pd-Cu contact can be found in Supplementary Fig. 1.

As can be seen in Fig. 2a, the reduction temperature for CuO is lowered by 30 °C when Pd is present on the same carbon nanotube. Pd is fully reduced before the start of the TPR experiment. Therefore, this reduction temperature shift is associated with CuO and indicates dissociated hydrogen from the Pd is spilling over onto the carbon nanotubes and reaching the CuO, causing the CuO to reduce at a lower temperature than normal. Comparable shifts in reduction temperature have been observed when Pd and CuO are in direct contact in the absence of the carbon nanotube bridge[16]. The observed changes in reducibility of CuO demonstrate that hydrogen can travel along the length of the carbon nanotube to reach the oxide and facilitate its reduction.

X-ray absorption spectroscopy was performed to study the oxidation state of copper for the Pd/CNT/CuO samples after various reductive treatments. For comparison, a sample of CuO powder was also physically mixed with nanotubes (CuO + CNT) to verify the lack of reduction of a CuO sample at similar degrees

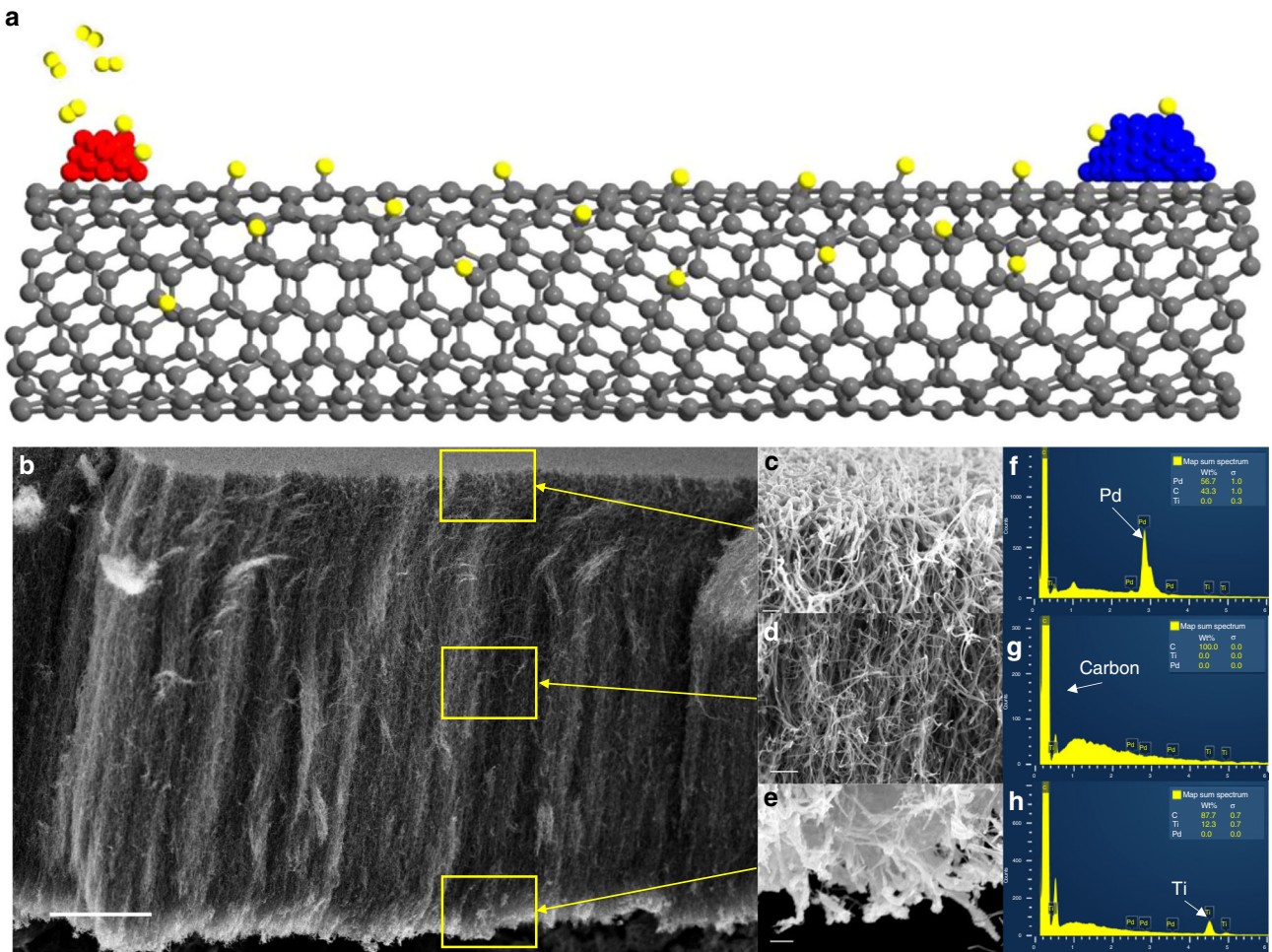

**Fig. 1** Metals and metal oxides deposited on opposing ends of a carbon nanotube. **a** Schematic depicting a metal (red) capable of dissociating hydrogen (yellow) onto a carbon nanotube where hydrogen can travel across to a metal oxide (blue). **b** SEM image of a nanotube forest with Pd and $TiO_2$ deposited on opposite ends through metal evaporation and after treatment in hydrogen for 1 h at 400 °C. (Scale bar in **b** indicates 15 micrometers). **c–e** Portions of the top, middle and bottom of the forest, respectively, at increased magnification. (Scale bar indicates from top to bottom 200, 500, and 250 nanometers). **f–h** EDS spectra corresponding to the locations indicated in **c–e**

of pretreatment in the absence of Pd. The initial oxidation state of the copper for the Cu containing materials both with (Fig. 2c) and without (Fig. 2d) Pd indicate that no metallic copper is initially present (Supplementary Tables 2 and 3). Mild reduction of the Pd/CNT/CuO showed a change in oxidation state upon treatment in the presence of a dilute hydrogen stream, whereas no change was observed in the CuO + CNT case up to 200 °C in dilute hydrogen. It should be noted that this CuO + CNT sample exhibits a similar reduction profile to the Cu/CNT sample discussed above (Supplementary Fig. 7). Further increasing the temperature for the Pd/CNT/CuO sample resulted in complete conversion to metallic copper. This confirms that hydrogen is spilling over from the Pd across the nanotube to the CuO. The same technique was attempted on the palladium assisted titania, with results shown in Supplementary Figs. 8–13 and Supplementary Table 4, but the small weight percentage of $TiO_2$ (0.65 wt% of the sample) coupled with the low fraction of $Ti^{3+}$ that results on the surface is below the detection limit of the instrument. This is consistent with reports by Rekoske and Barteau[17], where the degree of $Ti^{3+}$ generated after high temperature metal-assisted reduction of $TiO_2$ was only 0.09 wt% of the $TiO_2$ as $Ti^{3+}$.

**Catalytic reactions**. In order to demonstrate the capability of this approach to create promoter effects via continuous reduction of

an oxide across a carbon nanotube bridge, a mixed stream of acetic acid (AceOH) and furfural (FAL) was introduced to a catalyst containing Pd and $TiO_2$. AceOH ketonization is a reaction that is known to occur over $TiO_2$, with the rate of ketonization to form acetone (ACE) scaling with the number of exposed cations ($Ti^{3+}$ sites) on the surface[18,19]. While the reaction rate is proportional to the number of exposed cations or oxygen vacancies, these vacancies are not consumed in the process. This makes this reaction an ideal probe of the number of defects on the $TiO_2$ surface.

Cleavage of oxygen from carbonyl-containing species has been proposed over reducible oxide catalysts via a reverse Mars-van Krevelen mechanism[20]. Specifically over $TiO_2$, the selective conversion of FAL to form methylfuran has been proposed to occur over oxygen vacancies on the surface of $TiO_2$ that are produced by hydrogen spillover from the metal[21]. Other groups have proposed that the enhanced activity for methylfuran production is due to the metal Pd-$TiO_2$ interface based on rate and selectivity shifts observed after coating Pd with a porous $TiO_2$ film[22]. The selective deoxygenation of furfural has been a subject of many recent research articles due to its prevalence in biomass-derived streams[18,22–24]. This reaction would consume defects, requiring hydrogen to spillover from the Pd to re-reduce the oxide and complete the catalytic cycle. By co-feeding AceOH with FAL over a catalyst with physically separated sites, one can test

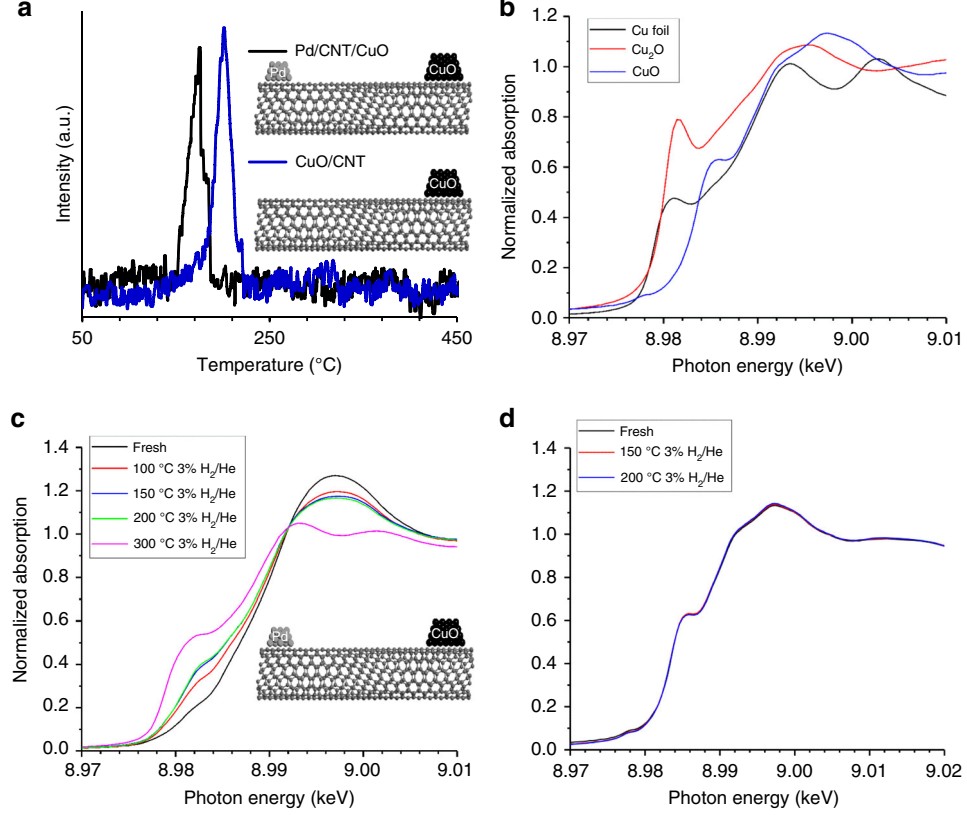

**Fig. 2** Facilitated reduction of Cu by Pd on the opposing end of a carbon nanotube. **a** TPR profiles of CuO/CNT and Pd/CNT/CuO catalysts performed with a flowing 5% $H_2$/Ar stream at a ramp rate of 5 °C/min. **b** Cu K edge XANES of Cu foil (black), $Cu_2O$ (red), and CuO. (blue). **c** XANES of fresh Pd/CNT/CuO (black) and after reduction in 3% $H_2$ in He at 100 (red), 150 (blue), 200 (green), and 300 °C (pink). **d** XANES of fresh CuO physically mixed with CNTs (black) and after reduction in 3% $H_2$ in He at 150 (red) and 200 °C (blue)

the significance of this path under reaction conditions. As can be observed in Fig. 3, the rate of acetone production from acetic acid is markedly enhanced in the presence of Pd on the other side of the nanotube (Pd/CNT/$TiO_2$) when compared with the catalyst containing only $TiO_2$ on the CNT, ($TiO_2$/CNT). This reveals that the Pd is capable of generating defects on the $TiO_2$ support necessary to enhance the rate of the ketonization reaction.

In addition to generating active sites, it should be noted that the catalyst with the separated Pd sites exhibits increased stability. One possible explanation for this behavior is that the low stability over the $TiO_2$/CNT catalyst could be due to furfural deoxygenation to yield methylfuran, which would consume oxygen vacancies in the process. This deactivation is even more pronounced if the $TiO_2$ contains several oxygen vacancies prior to introducing the reactants. This result can be found in Supplementary Fig. 2, where a severely prereduced $TiO_x$ exhibits similar rates of acetone formation to the Pd/CNT/$TiO_2$ catalyst after 40 min on stream, but the deactivation rate is clearly more pronounced when Pd is absent from the nanotube. These results indicate that if furfural deoxygenation to form methylfuran is occurring on $TiO_2$ defects, the regeneration of active sites by spillover from the Pd is fast enough to maintain a stable level of catalytic activity (oxygen vacancies) necessary for steady acetone production. In other words, defect regeneration is not rate limiting, or one would expect acetone formation rates to approach levels representative of a non-prereduced catalyst. An alternative explanation could be that side reactions such as aldol condensation and eventual coke formation lead to catalyst deactivation when furfural is present. In this case as well, the conclusion that Pd levels are sufficient to maintain a steady population of surface defects necessary for the proposed furfural

deoxygenation due to defects on the support under reaction conditions is still valid.

Since defects are present on the $TiO_2$ surface at steady state when the Pd and $TiO_2$ catalysts are separated on a nanotube bridge, the role of direct contact between the $TiO_2$ and the Pd nanoparticle may be investigated. Figure 4 shows that Pd on nanotubes alone (Pd/CNT) yields similar rates and selectivity to furan vs. methylfuran when compared with Pd and $TiO_2$ deposited on opposite ends of carbon nanotubes (Pd/CNT/$TiO_2$). Similar behavior is observed across a physical mixture of Pd/CNT and $TiO_2$/CNT, as can be seen in Supplementary Figs. 4 and 5. The small amount of lights observed, which consist of ring opening and CO-hydrogenation products, do not exhibit a trend. In contrast, selectivity for methylfuran increased significantly, when Pd is deposited on the same side as $TiO_2$ on the carbon nanotubes (Pd/$TiO_2$/CNT). It should be noted that furfural conversions were comparable over all catalysts (within 3%), and similar deactivation profiles with respect to furfural conversion were observed over all of the Pd containing catalysts as can be seen in Supplementary Fig. 6. These results suggest that active sites for methylfuran production are due to direct contact between the Pd and the $TiO_2$, as opposed to longer range defects as some have claimed[21]. In the case of furfural deoxygenation over $TiO_2$ supported catalysts, the selective C–O cleavage results from new sites generated via direct contact between the metal and the support rather than promoter effects.

A second set of probe reactions was performed to demonstrate the ability of this technique to probe promoter effects, where the primary role of the metal is to provide active sites on the support. Methoxy group conversion on $TiO_2$ supported catalysts has been suggested to be dependent on the population of $Ti^{3+}$ sites to yield phenolics and transalkylation products[25]. In this case, anisole was

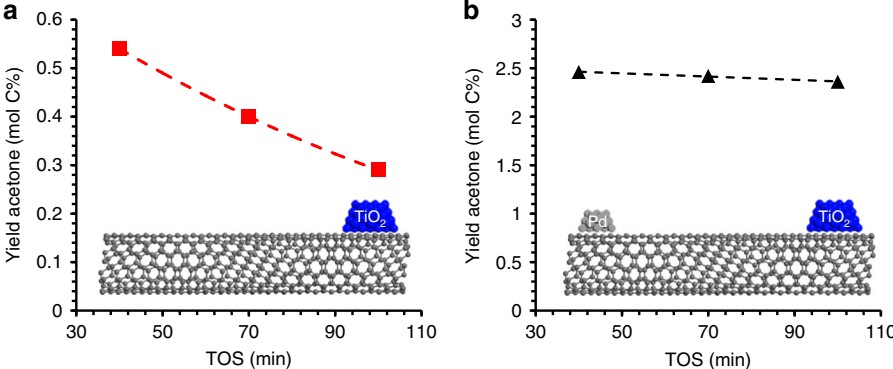

**Fig. 3** Evidence for stable Ti$^{3+}$ sites under reaction conditions when Pd is on nanotube. **a** Acetone yield when co-feeding furfural and acetic acid over TiO$_2$/CNT. **b** Acetone yield when co-feeding furfural and acetic acid under identical conditions over Pd/CNT/TiO$_2$. Reactions were both measured at $T = 400$ °C and $P = 1$ atm H$_2$ flow as a function of time on stream (TOS)

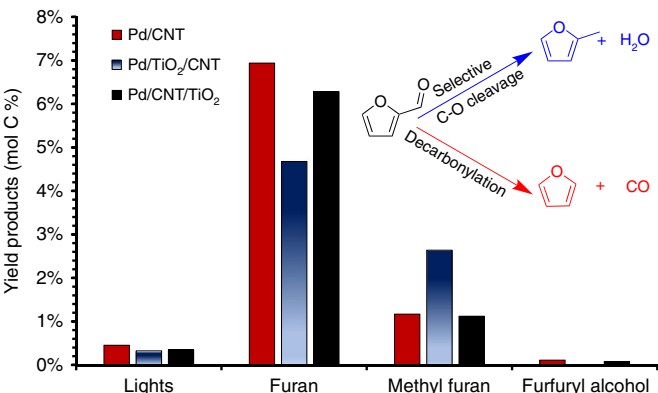

**Fig. 4** Furfural conversion over Pd and TiO$_2$ catalysts supported on CNTs. $T = 400$ °C, $P = 1$ atm, under flowing H$_2$, 30 min time on stream (TOS). The "lights" represent the sum of ring opening products and products resulting from the hydrogenation of CO found in low abundance. The inset to the upper right indicates the two primary competing reaction pathways to form furan or methylfuran

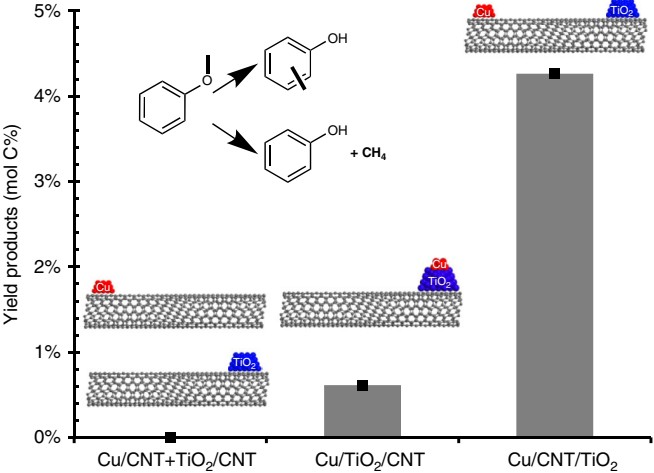

**Fig. 5** Anisole conversion over catalysts containing Cu and TiO$_2$. Sum of quantifiable products (cresol + phenol) resulting from anisole conversion over Cu and TiO$_2$ catalysts supported on CNTs. $T = 400$ °C, $P = 1$ atm, under a H$_2$ flow, 100 min TOS. The inset in the upper left indicates the main conversion products resulting from anisole under these conditions to generate cresol or phenol

chosen as the probe reaction to investigate the influence of promoter effects vs. sites located at the metal/support interface. Copper and titania were chosen as the metal/oxide combination in this case. Copper was chosen as the metal over palladium in this case due to copper's limited activity for hydrogenation and C–O scission reactions under these conditions, allowing the activity resulting from the defects on the TiO$_2$ support to be catalytically relevant. It should be noted that, unlike in the spillover case presented above, metallic Cu at 400 °C is capable of dissociating H$_2$. Three different Cu-TiO$_2$ catalysts were prepared, one with the metal subsequently deposited on the same side of the nanotube as the oxide (Cu/TiO$_2$/CNT), one with the two catalytic materials separated on the same nanotube (Cu/CNT/TiO$_2$), and a physical mixture of Cu supported on nanotubes (Cu/CNT) with TiO$_2$ supported on nanotubes (TiO$_2$/CNT).

Figure 5 shows the results for the three tested catalysts under steady state conditions. In the cases where spillover from the copper to the titania is possible through either direct contact or the carbon nanotube bridge, cresol and phenol are observed as products. The yield to individual products over these catalysts can be found in Supplementary Fig. 3. This likely results from an initial C–O cleavage to form surface methyl groups on the TiO$_2$ surface followed by subsequent alkylation of phenolic species to form cresol or interaction with hydrogen to form lights and phenol. A small amount of unquantifiable lights was also produced, containing

methane and methanol that are produced in conjunction with phenol. It is clear from Fig. 5 that anisole conversion products are only prevalent over catalysts whose population of Ti$^{3+}$ sites are maintained by the presence of a metal on the same nanotube.

Interestingly in the case of copper and titania separated on different ends of the same nanotube, the yield of products is much higher than in the case where Cu is sequentially deposited over Ti. This decrease is caused by the copper covering a portion of the titania, and therefore diminishing the number of exposed Ti$^{3+}$ sites. This demonstrates that not only does hydrogen spillover across the nanotube, but that spillover rates are sufficient to maintain the population of Ti$^{3+}$ sites necessary for sustained activity. This example shows that this experimental technique can be applied to investigate promoter effects as well.

## Discussion

The experimental approach described here can be used with a variety of different bifunctional catalysts to isolate the role of sites induced by direct contact from promoter effects. To illustrate the broad-ranging applications of this method, examples are given for both carbonyl group conversion at the metal-support interface as

well as carboxylic acid ketonization and methoxy group conversion on defects present on the support itself.

## Methods

**Materials**. Chemicals used for multi-walled carbon nanotube growth were isopropanol, iron nitrate nonahydrate, cobalt nitrate hexahydrate, aluminum nitrate nonahydrate, and 2-hydroxyethyl cellulose ($M_w \sim$ 1,300,000). All of the chemicals used were purchased from Sigma Aldrich. Silicon wafers of n-type were purchased from Wafer World, Inc. (SKU: 1186). In all, 18 MΩ water was obtained from an in-house filtration system and used in this study.

**Vertical multi-walled carbon nanotube growth**. Vertical multi-walled carbon nanotubes (VMWNTs) were grown by spin coating a catalyst solution on silicon wafers. First silicon wafers were cut using a diamond scribe into 22 mm x 22 mm square pieces. A catalyst solution was made containing 1.11 wt% iron nitrate nonahydrate, 0.39 wt% cobalt nitrate hexahydrate, 1.23 wt% aluminum nitrate nonahydrate, and 0.74 wt% 2-hydroxyethyl cellulose all with respect to water. The solution was spin coated on the silicon wafers by putting one millimeter of solution on the silicon wafer and spin coating using two stages, which followed one another. The silicon wafer was first spin coated at 500 rpm for 10 s and then the spin speed increased to 2000 rpm for 30 s. The silicon wafers with the solution were then allowed to dry overnight and calcined the next day.

The silicon wafer spin coated with catalyst solution was calcined the next day by placing the sample in a one-inch quartz diameter tube and connecting one end to an inlet line and the other an outlet line. The one-inch quartz diameter tube was placed in a furnace oriented horizontally. With a continuous flow of 150 sccm of air through the quartz tube the furnace was ramped to 450 °C at 10 °C per minute and then held at 450 °C for 2 h. After heating at 450 °C the reactor was allowed to cool to room temperature and the sample removed from the quartz tube.

For the growth of vertically aligned MWNTs the silicon wafer with catalyst was placed in a one-inch quartz diameter tube and connected to inlet and outlet gas lines. The quartz tube was placed in a furnace oriented horizontally for heating. With a flow of 300 sccm of hydrogen passing through the quartz tube the furnace was heated to 650 °C at 10 °C per minute and then held at 650 °C for 30 min. The flow of hydrogen was stopped and a flow of 300 sccm of argon was flowed through the quartz tube and the quartz tube ramped to a reaction temperature of 675 °C at a rate of 10 °C per minute. Then the flow of argon was changed to 200 sccm and flowed with ethylene at 200 sccm for 20 min. After the reaction the flow of ethylene was stopped and argon continued to flow through the quartz tube as the temperature decreased to room temperature.

**Depositing different catalyst on opposite ends of VMWNTs**. To facilitate the removal of the VMWNTs from the silicon wafer the sample was heated in air to help weaken the interaction between the VMWNTs and catalyst particles on the silicon wafer[26,27]. The sample was loaded into a one-inch diameter quartz tube and the quartz tube connected to inlet and outlet airlines. Next the quartz tube was loaded into a furnace oriented horizontally. Air was flowed through the quartz tube at 150 sccm while the furnace was heated to 480 °C at 10 °C per minute and then held at 480 °C for 2 h. After this step the sample was removed from the quartz tube once the temperature of the furnace reached room temperature.

Physical vapor deposition of palladium and titanium onto VMWNTs was completed by using a custom built vacuum evaporator. To evaporate metal onto the VMWNTs, the VMWNTs were placed in the vacuum evaporator sample area. Next a tungsten wire, 1 mm in diameter, was connected to two brass electrodes, which were connected to the power supply. Titanium wire, with a diameter of 0.050 cm, palladium wire, with a diameter of 0.025 cm, or copper wire, with a diameter of 0.01 cm, was wrapped around the tungsten wire. For all metals a length of 2 cm was wrapped around the tungsten wire. A quartz crystal monitor was used to determine the amount of titanium, copper, or palladium deposited on the VMWNTs. For making the catalyst where the palladium and titanium are separated palladium was first deposited on the VMWNTs. After evaporation of the palladium the side of the VMWNTs with palladium was attached to aluminum tape, which contains a carbon adhesive, to remove the VMWNTs from the silicon wafer. Following this the aluminum tape with VMWNTs was placed back into the vacuum evaporator sample area with the end of the VMWNTs without palladium face up. Titanium was then evaporated resulting in deposition on the other end of the VMWNTs. After evaporation of the titanium the VMWNTs on the aluminum tape the edges of the VMWNTs on the aluminum tape were cut off using a razor blade. This removal of the edges was performed to ensure that any palladium, which came into contact with titanium was removed since the edges of the VMWNTs on the aluminum tape could have both catalysts. Next the VMWNTs on aluminum tape were placed in a petri dish filled with isopropanol and soaked for 1 h to solubilize the carbon adhesive holding the VMWNTs to the carbon tape. After soaking for 1 h, the aluminum tape was shaken to help dislodge the VMWNTs from the carbon tape. The VMWNTs were then recovered from the isopropanol. To oxidize the titanium to titania or copper to copper oxide the VMWNTs with catalyst were placed in a one-inch quartz diameter tube and connected to inlet and outlet gas lines. The quartz tube was placed in a furnace oriented horizontally for heating. With a flow of 100 sccm of air passing through the quartz tube the furnace was heated to 350 °C at a rate of 10 °C per minute and then held at 350 °C for 60 min. This was done to ensure complete

oxidation of titanium to titania and copper to copper oxide[28–31]. Weight percentages of all deposited metals can be found in Supplementary Table 1.

**Characterization**. Scanning electron microscopy of the catalyst was performed using a Zeis Neon 40 EsB scanning electron microscope operating at an accelerating voltage of 5 kV for imaging and 10 kV when performing energy-dispersive X-ray spectroscopy. Transmission electron microscopy was performed by using a JEOL 200 FX equipped with a LaB$_6$ filament and operating at 200 kV.

TPR of the catalysts was carried out using an in house built system. An SRI 110 thermal conductivity detector (TCD) was used to analyze the effluent gas that was passed over a bed of Drierite® (anhydrous calcium sulfate) before entering the TCD, which was then analyzed with 5% hydrogen in argon mixture gas flown at the same rate. A flow rate of 30 sccm of 5% hydrogen in argon was passed through a ¼" quartz tube packed with quartz wool and 10 mg of sample. The quartz tube was mounted vertically in a furnace for heating. The temperature was ramped to 800 °C at 5 °C/min and then held at 800 °C for 10 min.

X-ray absorption spectra were collected on the bending magnet beamline of the Materials Research Collaborative Access Team (MRCAT) at the Advanced Photon Source, Argonne National Laboratory. Measurements were performed at the Cu K (8.979 keV) and Ti K (4.966 keV) edges in step-scan transmission mode. Catalyst samples were gently packed into a stainless-steel sample holder with ½ lb of force to prevent breakage of the nanotubes or compression of the separated metals. The sample holder was placed in a quartz reactor tube, sealed with Kapton windows by two Ultra-Torr fittings, and through which gases could be flowed. Measurements on the fresh samples were performed at room temperature after purging the reactor with He. A 3% H$_2$/He mixture flowing at 100 ccm was used to reduce the samples, which were held at the specific treatment temperatures for 30 min. Following reduction, the reactor was purged with He as it was cooled to room temperature and measurements were performed at room temperature under the inert atmosphere.

XAS data was processed using WinXAS v3.2 and standard normalization and background subtraction procedures[32]. For the Ti K edge, the oxidation state of Ti was determined by comparison of the sample edge energy to Ti foil, TiO$_2$, and Ti$_2$O$_3$ reference compounds. A least-squares fit in R-space of the $k^2$-weighted Fourier transformed EXAFS was performed to obtain coordination numbers and bond distances. The anatase phase of TiO$_2$ (6 Ti–O bonds at 1.96 Å) was used as an experimental reference for backscattering phase and amplitude fitting functions. A similar procedure was used to determine with EXAFS coordination parameters at the Cu K edge and experimental phase and amplitude fitting functions were constructed with NiO (6 Ni–O bonds at 2.09 Å) and Cu foil (12 Cu–Cu bonds at 2.55 Å). The quantities of Cu(II) and Cu(I) in each sample were estimated from the fractional coordination numbers obtained from fitting (i.e., the fitted value is the weighted average of the coordination numbers of Cu(II) (4 Cu–O bonds) and Cu(I) (2 Cu–O bonds) present in the sample).

TGA experiments were performed with the Cu/CNT samples described above as well as CuO powder (Aldrich 203130) physically mixed with nanotubes in a Netzsch STA 449F1 equipped with a pin thermocouple and a Netzsch nanobalance. Outlet gases were analyzed by mass spectroscopy on an Aeolos QMS. All data was taken with a correction file to account for gas viscosity changes during nonisothermal operation. Prior to pretreatment of the sample, the crucible was treated separately at 1000 °C under ultra-dry air followed by argon to ensure an accurate zero measurement. Zeroing of the scale was done at 100 °C under 40 sccm hydrogen to match the initial reduction conditions. The samples were pretreated by oxidation in air at 375 °C for 30 min as per the standard procedure for the catalyst to be used in reactions. Surface groups on the CNT were then removed by treatment for 1 h at 375 °C under 40 sccm/min of Ar. The samples were then cooled to 100 °C under inert Ar flowing at 20 sccm. Once the sample reached 100 °C, the gas was switch to a flow 40 sccm hydrogen until the mass stabilized from the gas switchover. The TPR was performed from 100 °C to 400 °C under 40 sccm hydrogen with a ramp rate of 3 °C/min.

**Chemical reactions**. Catalytic activity for the four different catalysts was tested in a quartz tube reactor (0.25 in OD) at atmospheric pressure and 400 °C. Catalyst particles were mixed with inert acid washed glass beads (Sigma Aldrich, Part number: G1277) with a particle size range of 212–300 μm and packed between two layers of quartz wool inside the reactor. The quartz tube was placed in a furnace oriented vertically and connected to an inlet gas line at the top and an outlet gas line at the bottom. The catalyst was reduced by flowing 100 sccm of hydrogen through the quartz tube and heating the furnace up to 400 °C and then holding at the same temperature for 1 h.

Anisole, distilled furfural (obtained from Sigma Aldrich; distilled and stored at −15 °C), and/or acetic acid were introduced to the reactor as noted experiments at a flow rate of 0.16 ml/h for anisole, 0.1 ml/h for furfural, and 0.22 ml/h for acetic acid. Ten milligram of catalyst was added in each case (with the exception of the physical mix samples, where 20 mg of material was added to the reactor to maintain consistent Pd, Ti, and Cu loadings). The outlet stream of the reactor was heated to 250 °C to prevent condensation of compounds in the transfer lines and then flowed through a six-port valve to allow for injection into a gas chromatography unit equipped with a flame ionization detector, Agilent 6890, using a HP-INNOWAX column (30 m, 0.25 μm) for product quantification. Identification of products was confirmed using a Shimadzu QP-2010 GCMS and

standards were used to quantify the various products in the flame ionization detector.

## Data availability

The data that support the findings of this study are available from the corresponding author on reasonable request.

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

## Acknowledgements

We acknowledge financial support from the National Science Foundation, Grant CAREER1653935. Use of the Advanced Photon Source is supported by the U.S. Department of Energy, Office of Science, and Office of Basic Energy Sciences, under Contract DE-AC02-06CH11357. MRCAT operations are supported by the Department of Energy and the MRCAT member institutions. E.C.W. and J.T.M. were supported in part by Center for Innovative Transformation of Alkane Resources (CISTAR) by the National Science Foundation under Cooperative Agreement No. EEC-1647722.

## Author contributions

S.C. conceived the idea for this research. N.B. constructed the metal evaporation apparatus and developed the selective deposition technique. Portions of this manuscript were adapted from N.B.'s Ph.D. dissertation[33]. N.B. and L.B. synthesized the carbon nanotube forests and developed the sample preparation and purification protocols for the various metals and carried out the SEM and EDX analysis of the samples. N.B. and L.H. carried out the TPR measurements. L.B. carried out the TGA measurements. L.H. carried out the furfural and acetic acid probe reactions. A.G. carried out the anisole probe reactions. J.M. coordinated the XAS measurements that were conducted by E.W. and L.B.. J.M. and E. W. analyzed the XAS data. All authors contributed to the analysis and interpretation of the data and participated in the writing of the manuscript.

## Additional information

**Competing interests:** The authors declare no competing interests.

