## [Peer Review File · Nature Communications]

Reviewers' comments:

Reviewer #1 (Remarks to the Author):

On the face of it this could be a very interesting paper but I feel it is over-hyped as it claim to show the nature of active sites for supported metal catalysts. I would expect many examples but in effect there are very few. I suggest they redraft the paper without the hype and discuss what they have with the one system. Then they need to show that their catalyst with the two metal particles is actually different from having two mixed catalysts one with one metal and one with the other. I suspect it will not be different. This is an important control. I suspect there are other key experiemnts that are needed.

Reviewer #2 (Remarks to the Author):

For the heterogeneous catalysis, understanding the location and nature of the catalytic active site is very important topic. In this present work, the authors try to discern the role of metal oxide via the controlling distance between Pd and TiO₂ with the carbon nanotubes. The method or design of experiment was interesting, however, it's very hard to verify the authors' discussion and conclusion with their data. Especially, the authors discussed the role or defects only based on their catalytic results, however there are no characterization results to support the claims. The TPR profile of the Pd-Cu/CNT catalyst was only evidence the authors provided, which is surely too weak and does not support the main hypothesis. Overall, obviously, it's too preliminary results to support the conclusion. Hence, I don't think it can be published on the Nature Communications, maybe available for the special journals.

In addition, the authors should correct many mistakes in their manuscript, such as the layout of Fig.1;

"Error! Bookmark not defined" in Page 3;

what's the lights and TOS in Fig.4.

Reviewer #3 (Remarks to the Author):

Crossley and co-workers have studied the effects of the spatial distribution of Pd and a metal oxide (CuO or TiO₂) on carbon nanotubes (CNT). To this end they have grown aligned CNT and then deposit Pd on one side and the oxide on the other side of the CNT 'forest'. This provides one with an interesting methodology to discriminate between the effects of, e.g., the direct contact of Pd and CuO on the one hand and Pd and CuO with micrometer distances on the other hand. This methodology is of general interest to nanomaterials and nanotechnology developments and might be of interest to the broad readership on Nature Communications. However, the current data set is too limited to claim (general) applicability. The data reported lack rigor and require substantial efforts to arrive at convincing conclusions. Some more specific comments are found below.

1. The authors refer to CNT as 'highways of hydrogen' without providing substantial evidence for this – vide infra.
2. The authors claim that they can control the CNT 'forest length' but do not report data on this. If they would provide variation of this length they may come up with effects of e.g. distance between Pd and CuO and relate this to reduction temperature or catalytic activity.
3. In Figure 1 scale bars are too vague to provide data on 'forest length'. Provide clear scale bars for all EM figures.
4. Figure 2. A modest effect of Pd on reducibility of CuO is apparent. Please quantify H₂ consumption and calculate degree of conversion of CuO to Cu.
5. Figure 3. Add data for Pd/CNT to sort out if synergy between Pd and TiO₂ is real or not.
6. Figure 4. Please report data with time-on-stream. See if stability is affected by presence of Pd.
7. Throughout the paper (except for figure 4) control experiments with Pd and TiO₂ on the same

side of the tubes are lacking. Should be added.

8. After metal deposition (Ti, Cu) full oxidation is claimed. Please verify with e.g. XPS, XANES.

9. From the CNT forest with metals deposited subsequently catalyst particles of 90-250 micrometer are produced. Provide details how this is done. This mechanical fracturing of the forest may bring about direct contact between Pd and metal oxides? Carefully describe and check.

10. For the catalysis experiments the authors describe the flow rate of reactants but do not provide catalyst amount. Add.

In summary, potentially interesting concept but paper needs extensive revision.

Reviewer #1

On the face of it this could be a very interesting paper but I feel it is over-hyped as it claim to show the nature of active sites for supported metal catalysts. I would expect many examples but in effect there are very few. I suggest they redraft the paper without the hype and discuss what they have with the one system. Then they need to show that their catalyst with the two metal particles is actually different from having two mixed catalysts one with one metal and one with the other. I suspect it will not be different. This is an important control. I suspect there are other key experiments that are needed.

We thank the reviewer for the comments. We have included another example where we report the conversion of anisole over Cu/TiO₂ catalysts, where promoter effects are responsible for the reaction chemistry, to broaden the scope of the paper. We feel that this example adds much balance to the paper, as it provides a contrasting example to the furfural case where the kinetically relevant sites are located at the metal/support interface. It should be pointed out that we study three systems in this paper, Pd/TiO₂, Cu/TiO₂, and Pd/Cu, with three probe reactions, furfural conversion, acetic acid conversion, and anisole conversion. We have also added a clarifying statement in the abstract to indicate that the paper includes these two contrasting examples to make it less general in an effort to avoid the feeling that the paper is over-hyped as the reviewer claims.

We have carried out the proposed physical mixture experiment for all three probe reactions-furfural conversion, acetic acid conversion, and anisole conversion. The results for furfural and acetic acid conversion indicate that Pd must be on the same nanotube as the TiO₂ for enhanced TiO₂ reduction rates to occur. In other words, the furan and methylfuran rates for the physical mix are identical to those presented in Figure 4 for Pd/CNT alone. The ketonization rate (and deactivation rate) observed with the physical mix is identical to that observed over TiO₂ alone in Figure 3. These results are presented in Supplementary Figure 4 and Supplementary Figure 5. The clear differences in these physical mix cases when compared with the Pd/CNT/TiO₂ increased ketonization rate and stability, as well as the Pd/TiO₂/CNT increased methylfuran production rates support the other data presented that show the important role of the nanotube for facilitating spillover. The third example for anisole conversion presented in Figure 5 also supports the argument of the crucial role that the nanotube plays on facilitating hydrogen spillover between the two catalytically active materials.

Physical mixture experiments (Pd/CNT + TiO₂/CNT) Note: Pd and TiO₂ amounts in the reactor are equivalent to those present in the Pd/CNT/TiO₂ case.

Supplementary Figure 4: Reaction feeding furfural over physical mixture Pd/CNT and TiO₂/CNT catalyst T= 400 °C, P = 1 atm, under a H₂ flow, 30 min TOS.

Supplementary Figure 5: Acetone yield when co-feeding furfural and acetic acid over physical mixture Pd/CNT and TiO₂/CNT At T= 400 °C, P = 1 atm, under a H₂ flow

Figure 5: Anisole conversion over catalysts containing Cu and TiO₂. Sum of quantifiable products (cresol + phenol) resulting from anisole conversion over Cu and TiO₂ catalysts supported on CNTs. T= 400 °C, P = 1 atm, under a H₂ flow, 100 min TOS. The inset in the upper left indicates the main conversion products resulting from anisole under these conditions to generate cresol or phenol.

Reviewer #2 (Remarks to the Author):

For the heterogeneous catalysis, understanding the location and nature of the catalytic active site is very important topic. In this present work, the authors try to discern the role of metal oxide via the controlling distance between Pd and TiO₂ with the carbon nanotubes. The method or design of experiment was interesting, however, it's very hard to verify the authors' discussion and conclusion with their data. Especially, the authors discussed the role of defects only based on their catalytic results, however there are no characterization results to support the claims. The TPR profile of the Pd-Cu/CNT catalyst was only evidence the authors provided, which is surely too weak and does not support the main hypothesis. Overall, obviously, it's too preliminary results to support the conclusion. Hence, I don't think it can be published on the Nature Communications, maybe available for the special journals. In addition, the authors should correct many mistakes in their manuscript, such as the layout of Fig.1; "Error! Bookmark not defined" in Page 3; what's the lights and TOS in Fig.4.

We thank the reviewer for pointing out the importance of the topic. While we feel that well thought out probe reactions and TPR experiments are quite convincing to explain and understand active sites, we certainly agree that the paper can be improved with additional characterization. In the current version, we have collaborated with Jeffrey Miller's group to carry out XAS experiments at Argonne National Labs, and the results support our claims. By carrying out in-situ measurements at Argonne National labs, we have confirmed the important role of the nanotube for spilling over hydrogen to reduce an oxide on the other end of the nanotube. This data is presented as Figure 3 in the current manuscript and is replicated here for convenience. By comparing Figure 3b and 3c, one can clearly observe the role of Pd on facilitating the reduction of the oxide CuO to Cu₂O and subsequently Cu metal at increasingly severe reductive treatments. We further used the technique to confirm the initial absence of metallic Cu in the starting material, as well as the presence of fully metallic Cu after reduction at 300°C. These experiments also allow us to determine particle size based on coordination number, which is presented in the manuscript as well.

We also express our appreciation for pointing out the errors in the manuscript. These have been corrected. We have also clarified the definition of TOS and added a description of "lights" in the figure captions.

The following text was added to the manuscript:

"X-ray adsorption spectroscopy was performed to study to oxidation state of copper for the Pd/CNT/CuO samples after various reductive treatments. For comparison, a sample of CuO nanoparticles was also physically mixed with nanotubes (CuO + CNT) to verify the lack of reduction of a CuO sample at similar degrees of pretreatment in the absence of Pd. The initial

oxidation state of the copper for the Cu containing materials both with (Figure 2c) and without (Figure 2d) Pd indicate that no metallic copper is initially present (Supplementary Table 3). Mild reduction of the Pd/CNT/CuO showed a change in oxidation state upon treatment in the presence of a dilute hydrogen stream, whereas no change was observed in the CuO + CNT case up to 200°C in dilute hydrogen. It should be noted that this sample CuO + CNT sample exhibits a similar reduction profile to the Cu/CNT sample discussed above (Supplementary Figure 7). Further increasing the temperature for the Pd/CNT/CuO sample resulted in complete conversion to metallic copper. This confirms that hydrogen is spilling over from the Pd across the nanotube to the CuO. The same technique was attempted on the palladium assisted titania, but the small weight percentage of TiO₂ (0.65 wt% of the sample) coupled with the low fraction of Ti³⁺ that results on the surface is below the detection limit of the instrument. This is consistent with reports by Rekoske and Barteau¹⁷ where the degree of Ti³⁺ generated after high temperature metal-assisted reduction of TiO₂ was only 0.09 wt% of the TiO₂ as Ti³⁺.”

Figure 2. Facilitated reduction of Cu by Pd on the opposing end of a carbon nanotube a) TPR profiles of CuO/CNT and Pd/CNT/CuO catalysts performed with a flowing 5% H₂/Ar stream at a ramp rate of 5°C/minute. b) Cu K edge XANES of Cu foil (black), Cu₂O (red) and CuO (blue). c) XANES of fresh Pd/CNT/CuO (black) and after reduction in 3% H₂ in He at 100 (red), 150 (blue), 200 (green), and 300 °C (pink). d) XANES of fresh CuO physically mixed with CNTs (black) and after reduction in 3% H₂ in He at 150 (red) and 200 °C (blue).

Reviewer #3 (Remarks to the Author):

Crossley and co-workers have studied the effects of the spatial distribution of Pd and a metal oxide (CuO or TiO₂) on carbon nanotubes (CNT). To this end they have grown aligned CNT and then deposit Pd on one side and the oxide on the other side of the CNT 'forest'. This provides one with an interesting methodology to discriminate between the effects of, e.g., the direct contact of Pd and CuO on the one hand and Pd and CuO with micrometer distances on the other hand. This methodology is of general interest to nanomaterials and nanotechnology developments and might be of interest to the broad readership on Nature Communications. However, the current data set is too limited to claim (general) applicability. The data reported lack rigor and require substantial efforts to arrive at convincing conclusions. Some more specific comments are found below.

1. The authors refer to CNT as 'highways of hydrogen' without providing substantial evidence for this – vide infra.

In addition to the TPR experiments and acetic acid ketonization probe reactions that were included in the prior draft, we have included several new experimental results that serve as evidence of hydrogen spillover across the nanotubes under reaction conditions. First, we have included physical mixture results that support our probe reaction experiments as discussed in our response to Reviewer 1 above. We have also included introduced a new set of reactions that rely on promoter effects, the conversion of anisole over Ti³⁺ sites, that is facilitated by spillover of hydrogen to the TiO₂ support to generate and sustain active sites where the enhancement in activity is not observed when the metal and oxide are not present on the same nanotube. We have also incorporated new XANES data for Pd and Cu separated on the same carbon nanotube, confirming the role of Pd at facilitating the reduction of CuO. These results are shown below:

Figure 2. Facilitated reduction of Cu by Pd on the opposing end of a carbon nanotube a) TPR profiles of CuO/CNT and Pd/CNT/CuO catalysts performed with a flowing 5% H₂/Ar stream at a ramp rate of 5°C/minute. b) Cu K edge XANES of Cu foil (black), Cu₂O (red) and CuO (blue). c) XANES of fresh Pd/CNT/CuO (black) and after reduction in 3% H₂ in He at 100 (red), 150 (blue), 200 (green), and 300 °C (pink). d) XANES of fresh CuO physically mixed with CNTs (black) and after reduction in 3% H₂ in He at 150 (red) and 200°C (blue).

AS discussed in the response to reviewer 2, the following text was added to the manuscript concerning the XAS measurements:

“X-ray adsorption spectroscopy was performed to study to oxidation state of copper for the Pd/CNT/CuO samples after various reductive treatments. For comparison, a sample of CuO nanoparticles was also physically mixed with nanotubes (CuO + CNT) to verify the lack of reduction of a CuO sample at similar degrees of pretreatment in the absence of Pd. The initial oxidation state of the copper for the Cu containing materials both with (Figure 2c) and without

(Figure 2d) Pd indicate that no metallic copper is initially present (Supplementary Table 3). Mild reduction of the Pd/CNT/CuO showed a change in oxidation state upon treatment in the presence of a dilute hydrogen stream, whereas no change was observed in the CuO + CNT case up to 200°C in dilute hydrogen. It should be noted that this sample CuO + CNT sample exhibits a similar reduction profile to the Cu/CNT sample discussed above (Supplementary Figure 7). Further increasing the temperature for the Pd/CNT/CuO sample resulted in complete conversion to metallic copper. This confirms that hydrogen is spilling over from the Pd across the nanotube to the CuO. The same technique was attempted on the palladium assisted titania, but the small weight percentage of TiO₂ (0.65 wt% of the sample) coupled with the low fraction of Ti³⁺ that results on the surface is below the detection limit of the instrument. This is consistent with reports by Rekoske and Barteau¹⁷ where the degree of Ti³⁺ generated after high temperature metal-assisted reduction of TiO₂ was only 0.09 wt% of the TiO₂ as Ti³⁺.”

We have also carried out new TGA experiments to compare our physical mixture of CuO + nanotubes with Cu deposited directly on nanotubes. The results are shown in the supplemental information:

Supplementary Figure 7: TGA in a stream of 66vol% hydrogen in argon at a ramp rate of 3°C/min.

This provides further evidence that the facilitated reduction of Cu in all cases is due to the presence of Pd on the same nanotube.

2. The authors claim that they can control the CNT ‘forest length’ but do not report data on this. If they would provide variation of this length they may come up with effects of e.g. distance between Pd and CuO and relate this to reduction temperature or catalytic activity.

We agree that this approach can be used to quantify the role of length on spillover rates across particles of varying distance. This could be measured if one were to use probe reactions that are in fact limited by hydrogen spillover from one catalytic species to the other. We feel that these experiments, although potentially very interesting, are better served in a second paper to avoid diluting the current message. We have removed the text that referenced forest length manipulation to avoid confusion in the current paper.

3. In Figure 1 scale bars are too vague to provide data on ‘forest length’. Provide clear scale bars for all EM figures.

We appreciate your feedback and have changed the scale bars to be more readable.

4. Figure 2. A modest effect of Pd on reducibility of CuO is apparent. Please quantify H₂ consumption and calculate degree of conversion of CuO to Cu.

The Cu is fully reduced from a mixture of 75% of the atomic atoms as CuO and 25% as Cu₂O, with full reduction to Cu metal after treatment to 300°C in dilute hydrogen as confirmed by XAS.

5. Figure 3. Add data for Pd/CNT to sort out if synergy between Pd and TiO₂ is real or not. We have conducted experiments with physical mixtures of Pd/CNT and TiO₂/CNT to support our hypothesis. Please see response to reviewer 1 for a more detailed discussion of the results obtained with the physical mixture.

6. Figure 4. Please report data with time-on-stream. See if stability is affected by presence of Pd.

This information is included in the supplemental information. While the ketonization activity is strongly influenced by the presence of the metal, the activity of the metal sites vs. TOS is similar for the two cases as shown below. This information was incorporated into the supplemental information as Supplementary Figure 6. The conversion of furfural does not occur in the absence of Pd.

Supplementary Figure 6: Furfural conversion (estimated based on disappearance of furfural) over Pd and TiO₂ catalysts supported on CNTs. T= 400 °C, P = 1 atm, under atmospheric H₂ flow.

7. Throughout the paper (except for figure 4) control experiments with Pd and TiO₂ on the same side of the tubes are lacking. Should be added.

In addition to the Pd/TiO₂/CNT sample shown in figure 4, we have included an updated example where sequential deposition of Cu/TiO₂/CNT samples were prepared are shown in Figure 5. In this case, it is shown that the sequential deposition of metal on top of the oxide can actually diminish the activity of the oxide in the case where promoter effects are present. For the TPR experiments, we cite literature examples of PdCu alloys to illustrate the shift in reduction temperature expected when the two materials were in direct physical contact.

8. After metal deposition (Ti, Cu) full oxidation is claimed.

After the metal is deposited on the nanotube it is first oxidized at 350°C for 1 hour. XAS also showed the titanium had completely reduced to Ti(IV) and the copper was 75% Cu(II) 25% Cu(I) with no residual metallic copper, see XAS on point 1 as well as the following graphs for the titania samples. These results are shown in the supplemental information Supplemental Figures 8-13 and Supplemental Tables 3-4. While small population of Ti³⁺ sites that are generated after subsequent surface reduction is below the detection limit, the initial oxidation state is confirmed with these measurements.

9. From the CNT forest with metals deposited subsequently catalyst particles of 90-250 micrometer are produced. Provide details how this is done.

This mechanical fracturing of the forest may bring about direct contact between Pd and metal oxides? Carefully describe and check.

We appreciate the insightful comment. The size of the catalyst particle, 90-250 micrometer, is not intentionally produced on a pelletizer, but rather when the forest is removed from the wafer or from the tape depending on the sample it comes off in pieces of this size. Carbon nanotubes grown in a vertical array are highly entangled as shown in figure 1. This entanglement keeps the nanotubes together until mild sonication is applied. For our purposes, it was not necessary to break up this entanglement, so sonication (which can also cause breakage of the nanotubes) was avoided. The nanotubes with deposited metals were left in their natural state once removed from the substrate. The statement that the reviewer is referring to has been removed the paper to avoid potential confusion.

The nanotubes were compressed slightly for the XAS, however due to their natural tendency to entangle they were able to stay in pellet form with less than ½ lb of pressure being applied to them. This is far less than the typical procedure followed for these experiments, and we believe that the agreement between the XAS results with our TPR results (were no sample compression was applied) supports the argument of no significant physical contact.

10. For the catalysis experiments the authors describe the flow rate of reactants but do not provide catalyst amount. Add.

We thank the reviewer for catching this. Each reaction was performed over 10 mg of catalyst, with the exception of the physical mixture where 20 mg of material was added to maintain Pd, Cu, and Ti loadings. These details have been added to the manuscript.

REVIEWERS' COMMENTS:

Reviewer #1 (Remarks to the Author):

It is clear that the authors have made a good attempt to address the questions raised and on the whole this is fine. I still feel the abstract and the introduction still try to show they are solving everything. The abstract does not help the reader and just needs to be factual as to what they have found with the catalysts investigated. It cannot be seen as a general discovery.

Reviewer #3 (Remarks to the Author):

The authors have taken the comments of the reviewers into account in their revised version. I support publication of the manuscript as is.

2nd round reviewers' comments:

Reviewer #1 (Remarks to the Author):

It is clear that the authors have made a good attempt to address the questions raised and on the whole this is fine. I still feel the abstract and the introduction still try to show they are solving everything. The abstract does not help the reader and just needs to be factual as to what they have found with the catalysts investigated. It cannot be seen as a general discovery.

We thank the reviewer for the comment. We have made several modifications to the manuscript to highlight the specific reactions that were studied. Within the abstract, we have added the following text:

“As illustrative cases, we use this approach to show that the selective transformation of furfural to methylfuran over Pd/TiO₂ occurs at the Pd-TiO₂ interface while anisole conversion to phenol and cresol over Cu/TiO₂ is facilitated by exposed Ti³⁺ cations on the support.”

Within the introduction section, we have added the following text:

“We use this approach to discern the kinetic relevance of these two families of potential active sites for the selective transformation of furfuraldehyde to methylfuran over a Pd/TiO₂ catalyst, as well as for the conversion of anisole to phenol and cresol over a Cu/TiO₂ catalyst. We confirm that the nanotubes are capable of serving as a shuttle to facilitate the generation of defects on the TiO₂ surface through a combination of temperature programmed reduction techniques and X-ray adsorption spectroscopy. We further use probe reactions that are sensitive to the number of exposed Ti³⁺ cations to confirm that said defects on the TiO₂ surface are present under reaction conditions required for the selective C-O cleavage reactions in question.”

We feel that these changes address the reviewers' concerns and help to add clarity to the manuscript by providing specific examples regarding the specific catalysts investigated and reactions studied. We would like to thank the reviewer for the constructive comments, the manuscript is certainly better as a result.

Reviewer #3 (Remarks to the Author):

The authors have taken the comments of the reviewers into account in their revised version. I support publication of the manuscript as is.

We would like to thank the reviewer for the helpful feedback. The manuscript is much improved as a result of your comments.